# Spiking Neural Network with Mixture of Heterogeneous Enhancement Experts for Underwater Object Detection

## Abstract

Underwater object detection faces unique challenges from haze, color distortion, and low contrast caused by light absorption and scattering, which significantly degrade image quality and detection performance. We propose HE-MoESNN, a spiking neural network that integrates a Mixture of Heterogeneous Enhancement Experts (HE-MoE) with a lightweight Forward Spiking Neural Network (FSNN) backbone. Unlike conventional MoE frameworks that feed identical inputs to all experts, HE-MoE assigns modality-specific inputs consisting of dehazing, color correction, and contrast enhancement to three parallel experts and fuses their outputs through a shared gating router. This design promotes expert diversity and enables the network to exploit complementary enhancement cues. FSNN improves efficiency by replacing costly ANN activations and conventional convolutions with signed spiking neurons and ternary convolutions, reducing computation while maintaining competitive accuracy. Extensive experiments on the RUOD and DUO benchmarks demonstrate that HE-MoESNN achieves state-of-the-art performance while maintaining high computational efficiency.

## 1 Introduction

Underwater object detection (UOD) is more challenging than generic object detection, as it not only suffers from common difficulties such as scale variation, occlusion, and class imbalance but also endures unique degradations caused by light absorption and scattering in water (Dai et al. (2024; 2023); Fu et al. (2023a)). These degradations manifest as color distortion, haze, and low contrast, which severely reduce image quality and impair detection accuracy. To mitigate these issues, underwater image enhancement is typically applied either as a pre-processing step or within joint learning frameworks that perform enhancement and detection simultaneously (Wang et al. (2024b)). Prior studies have identified color distortion, haze, and low contrast as primary factors that degrade detection performance. Motivated by these findings, we introduce a Mixture of Experts (MoE) strategy to incorporate multiple enhancement priors, aiming to improve the model's generalization across diverse underwater degradation conditions.

Mixture of Experts (MoE) models (Chen et al. (2022); Masoudnia & Ebrahimpour (2014) leverage multiple expert networks to learn diverse and complementary features, encouraging different experts to specialize in distinct input patterns (e.g., object scales, textures, or frequency ranges). Such specialization enables the model to build richer representations and generalize better to unseen data. Despite their success, MoE frameworks faces two key limitations: (i) in conventional MoE designs, all experts receive the same input representation (as illustrated in Fig. 1 (a)), often resulting in homogeneous experts that learn overlapping or redundant features. Without an explicit mechanism to enforce diversity, MoE fails to fully exploit complementary information, limiting potential performance gains. (ii) Even with computationally efficient Top-k routing (Zhou et al. (2022)), which activates only a small subset of experts during inference, the full set of experts must still be maintained in memory. This overhead increases memory usage and computational cost, making deployment challenging on resource-constrained underwater devices where efficiency is critical.

To address these challenges, we propose HE-MoESNN, a spiking neural network with a Mixture of Heterogeneous Enhancement Experts (HE-MoE) tailored for underwater object detection. The

framework introduces **two key innovations**: (i) a novel HE-MOE module that integrates multiple enhancement priors, with each expert explicitly designed to handle a specific enhancement modality. These priors target three major underwater degradation: color distortion, haze effects, and low contrast. The design encourages experts to specialize in distinct enhancement domains and capture diverse, complementary features. (ii) A lightweight expert architecture, where the conventional forward artificial neural network (FANN) is replaced by a forward spiking neural network (FSNN), reducing computational cost while preserving high detection accuracy.

## 2 RELATED WORK

### 2.1 UNDERWATER OBJECT DETECTION

Underwater images frequently suffer from color distortion, low contrast, and haze due to light absorption, scattering, and noise, which significantly degrade the performance of underwater object detection (Chen et al. (2024)). These degradations obscure object boundaries, suppress fine structural details, and reduce the discriminative power of learned features, making reliable detection particularly challenging in turbid or low-light conditions. Numerous studies have demonstrated that image enhancement serves as an effective pre-processing step for improving detection. Color correction methods (Lai et al. (2025); Sarkar et al. (2024)) restore natural colors by compensating for the loss of long-wavelength components caused by light absorption, making objects more visually distinguishable. Dehazing techniques (Uke et al. (2025); Mo et al. (2025)) suppress backscatter and enhance scene visibility, producing clearer images with more reliable features for detectors. Contrast enhancement approaches (Almutiry et al. (2024); Zhang et al. (2025)) emphasize object boundaries and local details, aiding detectors in accurately localizing and classifying targets, particularly in low-light or noisy conditions. Collectively, these enhancement techniques play a crucial role in restoring perceptual quality, enabling more robust feature learning, and ultimately improving the generalization of detection models across diverse and dynamically changing underwater environments.

### 2.2 SPIKING NEURAL NETWORK FOR OBJECT DETECTION.

Spiking Neural Networks (SNNs) have emerged as a promising solution for computationally efficient object detection, owing to their event-driven nature and inherently sparse computation (Yamazaki et al. (2022)). A critical factor influencing SNN performance is the choice of spiking neuron model, which determines how information is encoded and how temporal dynamics are captured. Integrate-and-Fire (IF) neurons (Jin et al. (2023); Qiu et al. (2023)) are widely adopted in SNN-based object detection due to their low computational cost, simple implementation, and good hardware compatibility. However, their binary output limits representational capacity and typically requires more timesteps to match ANN accuracy, resulting in higher latency. To overcome the representational limitations of IF neurons, signed spiking neurons (Kim et al. (2020); Miao et al. (2025)) have been introduced, generating spikes in $-1, 0, +1$ form to enhance feature expressiveness. This richer representation narrows the performance gap between ANNs and SNNs, allowing faster convergence and competitive performance with fewer timesteps. While SNNs have been applied to generic object detection, their potential for underwater object detection remain largely unexplored. The unique challenges of UOD, such as color distortion, haze, and low contrast, make it a compelling testbed for developing efficient, event-driven SNN solutions.

## 3 METHOD

This section outlines the proposed HE-MoESNN, details the HE-MoE module for incorporating diverse enhancement priors to boost detection performance, and introduces the FSNN backbone, which offers computational advantages over the conventional FANN backbone.

### 3.1 OVERVIEW OF HE-MOESNN FRAMEWORK

The proposed HE-MoESNN is a spiking neural network (SNN) derived from the ANN-based YOLOX architecture (Ge et al. (2021)). To alleviate the adverse effects of severe underwater image degradation on detection performance, the HE-MoE module is integrated before the YOLOX

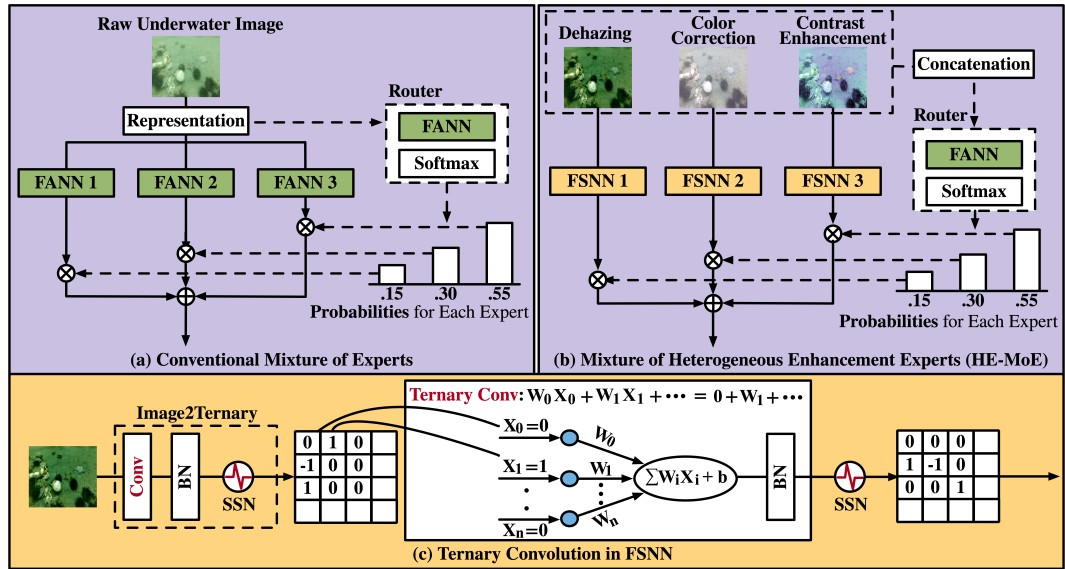

Figure 1: Comparison of (a) conventional MoE and (b) the proposed HE-MoE. In standard MoE, the same input can lead to homogeneous, redundant experts, while HE-MoE employs heterogeneous, complementary experts for better learning. (c) HE-MoE adopts FSNN as the expert architecture, where signed spiking neurons (SSN) convert convolutions into ternary form to reduce computation.

backbone. As illustrated in Fig. 1 (b), the HE-MoE module comprises three parallel enhancement experts: a dehazing expert (Meng et al. (2013)), a color correction expert (Jha & Bhandari (2024)), and a contrast enhancement expert (Zhang et al. (2023)). While these experts improves detection performance, they also increase computational overhead. To balance accuracy and efficiency, we adopt FSNN as the architecture for each expert, replacing conventional FANN activations with signed spiking neurons, as illustrated in Fig. 1 (c). Each enhanced modality is first converted from continuous-valued images to ternary-valued representations via an Image2Ternary module, followed by computationally efficient ternary convolutions. The overall loss function remains identical to that of YOLOX.

## 3.2 MIXTURE OF HETEROGENEOUS ENHANCEMENT EXPERTS (HE-MoE)

Conventional Mixture of Experts (MoE) models aim to learn diverse and complementary representations by combining the outputs of multiple experts. Given an input image representation $x$, a gating network $G(\cdot)$ produces a probability distribution over $N =$ experts $\{E_1, E_2, \ldots, E_N\}$, and the final output is computed as:

$$y = \sum_{i=1}^{N} G_i(x) E_i(x), \qquad \text{where} \quad \sum_{i=1}^{N} G_i(x) = 1. \tag{1}$$

In practice, however, conventional MoE often yields homogeneous experts with overlapping or redundant representations, since all experts receive the same input $x$. The absence of enforced diversity limits the ability of MoE to fully exploit complementary information.

To overcome this limitation, we propose the Mixture of Heterogeneous Enhancement Experts. Unlike conventional MoE, where all experts share the same input, HE-MoE employs modality-specific experts guided by prior domain knowledge. The input image $x$ is first processed by three dedicated enhancement experts: dehazing $E_d(\cdot)$, color correction $E_c(\cdot)$, and contrast enhancement $E_e(\cdot)$, to produce three modality-specific feature representations:

$$z_d = E_d(x), \qquad z_c = E_c(x), \qquad z_e = E_e(x). \tag{2}$$

A shared gating router then predicts the routing probabilities for each modality-specific expert:

$$G_d(x) + G_c(x) + G_e(x) = 1, \qquad y = G_d(x) z_d + G_c(x) z_c + G_e(x) z_e. \tag{3}$$

This formulation explicitly enforces expert specialization, ensuring that each expert is responsible for a specific type of underwater degradation (haze, color distortion, or low contrast). By incorporating these enhancement priors, HE-MoE learns complementary, domain-aware representations that lead to more robust and discriminative underwater feature extraction.

## 3.3 Comparison of Computational Cost between FANNs and FSNNs

The dominant computational cost in both FANNs and FSNNs arises from convolution operations. **Computational Cost of Convolution in FANNs.** In FANNs, the output activation of a convolutional neuron is computed as the nonlinear transformation of a weighted sum of its inputs. This process can be formally expressed as:

$$y = \varphi \left( \sum_{i=1}^{n} w_i x_i + b \right) \tag{4}$$

where $x_i \in \mathbb{R}$ are continuous-valued inputs, $w_i \in \mathbb{R}$ are learnable weights, $b \in \mathbb{R}$ is a bias term, $\varphi$ denotes a nonlinear activation function (e.g., SiLU), and $n$ represents the number of input connections. This operation yields continuous-valued outputs. Each neuron therefore performs $n$ multiplications and $n$ additions ($n - 1$ additions for the weighted sum and one for the bias). Thus, the computational cost per convolutional neuron in a FANN can be expressed as

$$C_{\text{ANN}} \propto n \cdot C_{\text{mul}} + n \cdot C_{\text{add}} \tag{5}$$

where $C_{\text{mul}}$ and $C_{\text{add}}$ are the costs of a single multiplication and addition, respectively.

**Computational Cost of Convolution in FSNNs.** In the proposed FSNN, the activation function $\varphi$ is replaced with a signed spiking neuron, resulting in ternary-valued outputs ($x_i \in \{-1, 0, +1\}$), as shown in Fig. 1 (c). Because $x_i$ takes ternary values, floating-point multiplications are avoided. When $x_i = +1$, the $i^{th}$ input contributes a simple addition of $+w_i$; when $x_i = -1$, it becomes a subtraction of $w_i$; and when $x_i = 0$, the computation is skipped entirely.

Let $p_0 = \Pr(x_i = 0)$ represent the input sparsity, i.e., the probability that a given input is zero. The expected number of arithmetic operations is therefore reduced to $(1 - p_0) \cdot n$, where $n$ is the number of input connections. Consequently, the computational cost per convolutional neuron in FSNNs is:

$$C_{\text{SNN}} \propto (1 - p_0) \cdot n \cdot C_{\text{add}}, \tag{6}$$

For a clear comparison, combining Eqs. 5 and 6 yields the relative computational cost ratio between FSNNs and FANNs:

$$\frac{C_{\text{SNN}}}{C_{\text{ANN}}} = \frac{(1 - p_0) \cdot C_{\text{add}}}{C_{\text{mul}} + C_{\text{add}}}, \tag{7}$$

This expression highlights that FSNNs achieve higher computational efficiency by simultaneously eliminating multiplications and leveraging input sparsity, resulting in significantly fewer arithmetic operations than conventional FANNs.

## 4 Experiments

### 4.1 Implementation Details

The proposed HE-MoESNN models are implemented using PyTorch in combination with Spiking-Jelly (Fang et al. (2023)), a widely used library for spiking neural network research. Inspired by YOLOX (Ge et al. (2021)), we construct three model variants of different scales: **HE-MoESNN-T** (Tiny), **HE-MoESNN-S** (Small), and **HE-MoESNN-L** (Large), allowing a systematic study of the trade-off between accuracy and computational cost. Training is performed using the Adam optimizer and a StepLR scheduler, an initial learning rate of 0.01, and a batch size of 16. The number of training epochs is scaled with model size (500 epochs for HE-MoESNN-L and 300 epochs for HE-MoESNN-T/S). All experiments are carried out on a workstation equipped with an Intel(R) Xeon(R) Silver 4114 CPU (2.20 GHz) and a Tesla V100 GPU with 32 GB of memory.

Table 1: The quantitative performance of representative detection frameworks on the RUOD dataset. The bold text represents the best performance, while the red text indicates the second-best.

| Methods | | Models | Backbones | Params | FLOPs | mAP | AP$_{0.50}$ | AP$_{0.75}$ | AP$_s$ | AP$_m$ | AP$_l$ |
|---|---|---|---|---|---|---|---|---|---|---|---|
| Non-Spiking | Generic | RepPoints | ResNet101 | 55.82M | 256.00G | 53.2 | 82.2 | 60.1 | 28.2 | 44.9 | 57.8 |
| | | FoveaBox | ResNet101 | 56.68M | 268.29G | 44.8 | 80.2 | 45.2 | 18.0 | 37.5 | 49.1 |
| | | ATSS | ResNet101 | 51.13M | 267.26G | 54.0 | 80.3 | 60.2 | 18.0 | 40.0 | 59.5 |
| | | DetectoRS | DResNet50 | 123.23M | 90.05G | 53.3 | 84.1 | 58.7 | 30.8 | 46.6 | 57.8 |
| | | YOLOv10 | CSPNet | 24.40M | 120.30G | 55.5 | 84.7 | 62.5 | 21.9 | 47.0 | 60.5 |
| | Underewater | BoostRCNN | ResNet50 | 45.95M | 54.71G | 53.9 | 80.6 | 59.5 | 11.6 | 39.0 | 59.3 |
| | | RFTM | ResNet50 | 75.58M | 91.06G | 53.3 | 80.2 | 57.7 | 11.8 | 39.2 | 59.3 |
| | | ERLNet | SiEdgeR50 | 45.95M | 54.71G | 54.8 | 83.1 | 60.9 | 14.7 | 41.4 | 59.8 |
| | | GCCNet | SwinFT | 38.31M | 78.93G | 56.1 | 83.2 | 60.5 | 11.7 | 41.9 | 62.1 |
| | | DJLNet | ResNet50 | 58.48M | 69.51G | 57.5 | 83.7 | 62.5 | 15.5 | 41.8 | 63.1 |
| Spiking | SNN | Spiking-YOLO | TinyYOLO | 23.10M | 136.9G | 49.8 | 80.7 | 55.1 | 17.6 | 42.5 | 54.4 |
| | | EMS-YOLO | EMSResNet34 | 14.40M | 37.00G | 52.0 | 82.9 | 58.4 | 19.3 | 44.2 | 57.4 |
| | | SpikingYOLOX | SNNCSPNet | 49.53M | 151.69G | 57.0 | 84.2 | 61.2 | 11.0 | 41.2 | 63.0 |
| | Ours | HE-MoESNN-T | MoESNN-T | 5.19M | 22.28G | 56.1 | 83.7 | 60.4 | 13.0 | 41.1 | 62.0 |
| | | HE-MoESNN-S | MoESNN-S | 8.56M | 35.86G | 57.1 | 84.3 | 61.3 | 15.7 | 42.8 | 63.1 |
| | | HE-MoESNN-L | MoESNN-L | 51.37M | 170.03G | 59.0 | 85.5 | 64.0 | 13.7 | 44.2 | 65.0 |

Table 2: The quantitative performance of representative detection frameworks on the DUO dataset. The bold text represents the best performance, while the red text indicates the second-best.

| Methods | | Models | Backbones | Params | FLOPs | mAP | AP$_{0.50}$ | AP$_{0.75}$ | AP$_s$ | AP$_m$ | AP$_l$ |
|---|---|---|---|---|---|---|---|---|---|---|---|
| Non-Spiking | Generic | RepPoints | ResNet101 | 55.82M | 256.00G | 59.4 | 80.4 | 70.1 | 55.5 | 59.6 | 60.1 |
| | | FoveaBox | ResNet101 | 55.68M | 268.29G | 53.7 | 78.4 | 63.9 | 55.3 | 54.3 | 54.6 |
| | | ATSS | ResNet101 | 51.13M | 267.26G | 55.4 | 79.2 | 63.2 | 55.7 | 55.7 | 56.0 |
| | | DetectoRS | DResNet50 | 123.23M | 90.05G | 58.9 | 81.4 | 68.3 | 49.6 | 57.6 | 61.8 |
| | | YOLOv10 | CSPNet | 24.40M | 120.30G | 62.3 | 84.7 | 70.9 | 48.5 | 63.9 | 61.6 |
| | Underewater | BoostRCNN | ResNet50 | 45.95M | 54.71G | 53.9 | 80.6 | 59.5 | 11.6 | 39.0 | 59.3 |
| | | RFTM | ResNet50 | 75.58M | 91.06G | 60.1 | 79.4 | 68.1 | 49.0 | 61.1 | 59.5 |
| | | ERLNet | SiEdgeR50 | 45.95M | 54.71G | 61.2 | 81.4 | 69.5 | 55.2 | 62.2 | 60.8 |
| | | GCCNet | SwinFT | 38.31M | 78.93G | 61.1 | 81.6 | 67.3 | 52.5 | 63.6 | 59.3 |
| | | DJLNet | ResNet50 | 58.48M | 69.51G | 65.6 | 84.2 | 73.0 | 55.6 | 67.4 | 64.1 |
| Spiking | SNN | Spiking-YOLO | TinyYOLO | 23.10M | 136.9G | 60.6 | 78.8 | 67.3 | 52.0 | 61.2 | 59.3 |
| | | EMS-YOLO | EMSResNet34 | 14.40M | 37.00G | 62.7 | 80.8 | 69.1 | 53.8 | 63.5 | 61.1 |
| | | SpikingYOLOX | SNNCSPNet | 49.53M | 151.69G | 64.3 | 81.9 | 70.4 | 54.7 | 65.0 | 63.5 |
| | Ours | HE-MoESNN-T | MoESNN-T | 5.19M | 22.28G | 65.1 | 83.8 | 72.0 | 55.6 | 66.0 | 64.0 |
| | | HE-MoESNN-S | MoESNN-S | 8.56M | 35.86G | 67.0 | 85.5 | 73.9 | 55.9 | 69.2 | 66.2 |
| | | HE-MoESNN-L | MoESNN-L | 51.37M | 170.03G | 68.0 | 86.6 | 74.1 | 55.5 | 69.3 | 67.1 |

## 4.2 Evaluation Datasets and Metrics

To benchmark the proposed HE-MoESNN framework, we adopt the RUOD dataset (Fu et al. (2023b)) and DUO (Liu et al. (2021)), two widely recognized benchmarks for underwater object detection. The evaluation metrics include mean Average Precision (mAP) across multiple IoU thresholds ($mAP@[0.5:0.05:0.95]$) and Average Precision at specific thresholds ($AP_{0.50}$ and $AP_{0.75}$). To assess performance across different object scales, we also report $AP_s$, $AP_m$, and $AP_l$ for small, medium, and large objects, respectively. In addition to detection accuracy, efficiency is evaluated in terms of computational cost (FLOPs) and model size (parameter count, Params).

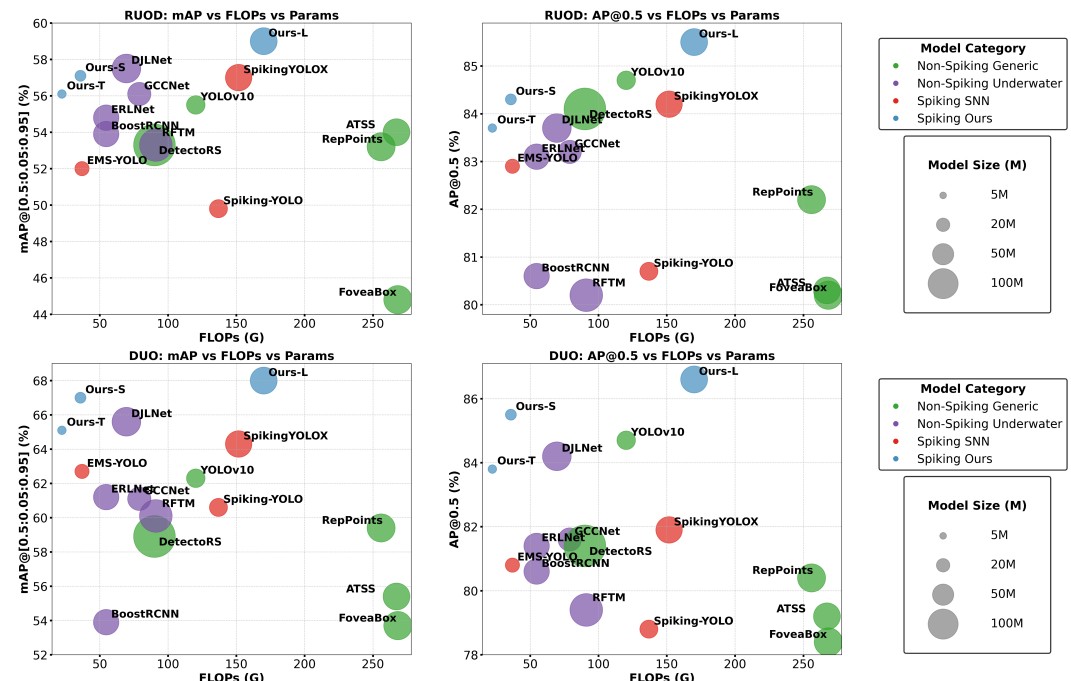

Figure 2: Precision ($AP@0.50$, $mAP@[0.50:0.05:0.95]$) and efficiency (FLOPs, Params) comparison of representative detection models on RUOD (top) and DUO (bottom).

## 4.3 COMPARISONS WITH STATE-OF-THE-ART METHODS

We compare HE-MoESNN against a diverse set of state-of-the-art object detectors, including five generic detectors (YOLOv10 (Wang et al. (2024a)), RepPoints (Yang et al. (2019)), FoveaBox (Kong et al. (2020)), ATSS (Zhang et al. (2020)), and DetectoRS (Qiao et al. (2021)), five leading underwater detectors (DJLNet (Wang et al. (2024b)), GCCNet (Dai et al. (2024)), ERLNet (Dai et al. (2023)), RFTM (Fu et al. (2023a)), and BoostRCNN (Song et al. (2023)), and three SNN-based detectors (SpikingYOLOX (Miao et al. (2025)), EMS-YOLO (Su et al. (2023)), and Spiking-YOLO (Kim et al. (2020)).

**Precision Analysis:** The proposed HE-MoESNN-L achieves state-of-the-art detection performance on both RUOD and DUO, as reported in Tables 1 and 2. On RUOD, DJLNet ranks second, benefiting from its image decolorization module, which effectively mitigates color distortions caused by light absorption. These results confirm the crucial role of image enhancement in underwater object detection. Unlike DJLNet, which focuses primarily on color correction, HE-MoESNN integrates three complementary enhancement techniques within a unified mixture-of-experts (MoE) framework. By jointly leveraging priors for dehazing, color correction, and contrast enhancement, HE-MoESNN delivers superior detection accuracy and robust feature representation. On DUO, the lightweight HE-MoESNN-S achieves the second-highest detection accuracy while offering the best computational efficiency, requiring only 8.56M parameters and 35.86G FLOPs. This favorable trade-off between accuracy and efficiency is largely due to the integration of multiple enhancement priors and the FSNN backbone, which collaboratively handle underwater image degradations in a computationally efficient manner.

**Efficiency Analysis:** Fig. 2 compares detection accuracy and efficiency metrics (FLOPs and Params) across representative detection models. The proposed HE-MoESNN-L achieves the highest precision on both datasets but comes with relatively large computational costs. In contrast, HE-MoESNN-S offers the second-best accuracy while achieving the highest efficiency on DUO, with only 8.56M parameters and 35.86G FLOPs, striking an excellent balance between precision and computational cost. On RUOD, although HE-MoESNN-S ranks third and marginally lags behind DJLNet in accuracy, it demonstrates a clear efficiency advantage, using significantly fewer re-

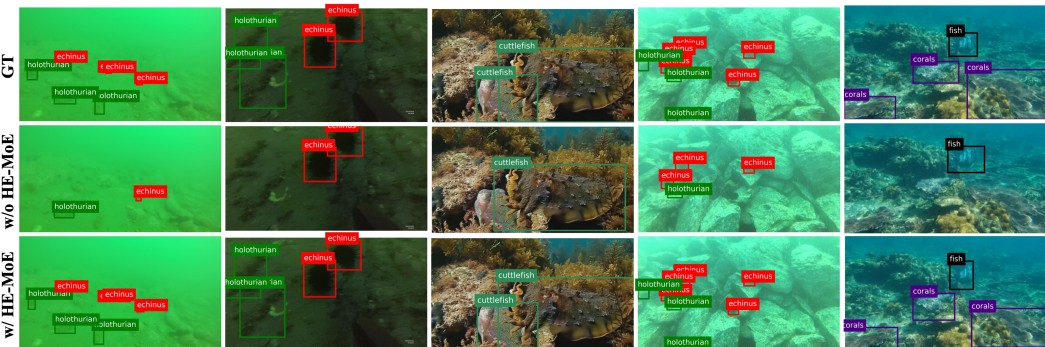

Figure 3: Detection result comparison of the proposed HE-MoESNN-L model with ($w$/) and without ($w$/$o$) the HE-MoE architecture. The $w$/$o$ HE-MoE model often fails under severe color distortion, heavy hazing, and low-contrast conditions, whereas the $w$/ HE-MoE variant consistently achieves accurate detections in these challenging scenarios.

Table 3: Performance comparison of frameworks with ($w$/) and without ($w$/$o$) HE-MoE, and with ($w$/) FANN versus FSNN backbones.

| Models | HE-MoESNN-T | | HE-MoESNN-S | | HE-MoESNN-L | |
|---|---|---|---|---|---|---|
| **HE-MoE** | *w/o* HE-MoE | *w/* HE-MoE | *w/o* HE-MoE | *w/* HE-MoE | *w/o* HE-MoE | *w/* HE-MoE |
| **mAP** | 52.6 | 56.1+3.5% | 54.0 | 57.1+3.1% | 57.0 | 59.0+2.0% |
| **AP$_{0.50}$** | 82.0 | 83.7+1.7% | 83.0 | 84.3+1.3% | 84.2 | 85.5+1.3% |
| **AP$_{0.75}$** | 56.2 | 60.4+4.2% | 57.7 | 61.3+3.6% | 61.2 | 64.0+2.8% |
| **FSNN** | *w/* FANN | *w/* FSNN | *w/* FANN | *w/* FSNN | *w/* FANN | *w/* FSNN |
| **mAP** | 57.4 | 56.1-1.3% | 57.5 | 57.1-0.4% | 59.2 | 59.0-0.2% |
| **AP$_{0.50}$** | 84.9 | 83.7-1.2% | 85.3 | 84.3-1.0% | 85.4 | 85.5+0.1% |
| **AP$_{0.75}$** | 61.9 | 60.4-1.5% | 62.0 | 61.3-0.7% | 64.5 | 64.0-0.5% |

sources (8.56M vs. 58.48M parameters and 35.86G vs. 69.51G FLOPs). This efficiency makes HE-MoESNN-S particularly suitable for real-time deployment in resource-constrained environments.

## 4.4 ABLATION STUDY

To validate the effectiveness of the proposed HE-MoESNN frameworks, we perform the ablation study on the RUOD dataset, focusing on two key components: the HE-MoE module and its individual enhancement experts.

### 4.4.1 ABLATION STUDY ON THE HE-MOE ARCHITECTURE

Two sets of experiments are conducted to assess the contribution of the HE-MoE architecture. First, we compare models trained with and without the HE-MoE module to quantify its direct impact on detection performance. Second, we evaluate the HE-MoE architecture with two different backbones: FANN and FSNN. This examines how backbone selection influences the overall performance.

The results are presented in Table 3, from which three key observations can be drawn. First, incorporating the HE-MoE module, which integrates dehazing, color correction, and contrast enhancement priors, substantially improves detection performance across all three model variants. This improvement highlights the importance of jointly addressing multiple forms of underwater image degradation, allowing the model to extract more reliable features and achieve better object localization under challenging conditions. Second, the enhancement priors provide a noticeably larger accuracy boost for the small model compared to the larger variants. This is likely because smaller models have limited representational capacity and struggle to handle severe underwater image degradation independently. By providing clearer, higher-quality inputs, HE-MoE helps close the performance gap between small and large models. Finally, the HE-MoE architecture with the FANN backbone outperforms its FSNN counterpart, indicating that FANN provides stronger feature extraction capa-

Table 4: Performance comparison of frameworks with (*w/*) the standard MoE and the HE-MoE.

| Models / Metrics | HE-MoESNN-T | | HE-MoESNN-S | | HE-MoESNN-L | |
|---|---|---|---|---|---|---|
| | *w/* MoE | *w/* HE-MoE | *w/* MoE | *w/* HE-MoE | *w/* MoE | *w/* HE-MoE |
| **mAP** | 54.3 | 56.1+1.8% | 55.7 | 57.1+1.4% | 57.8 | 59.0+1.2% |
| **AP$_{0.50}$** | 82.5 | 83.7+1.2% | 83.9 | 84.3+0.4% | 84.9 | 85.5+0.6% |
| **AP$_{0.75}$** | 58.1 | 60.4+2.3% | 59.4 | 61.3+1.9% | 62.9 | 64.0+1.1% |

Table 5: Performance comparison between HE-MoESNN variants with different enhancement modalities: dehazing (DH), color correction (CC), and contrast enhancement (CE).

| DH | CC | CE | HE-MoESNN-T | | HE-MoESNN-S | | HE-MoESNN-L | |
|---|---|---|---|---|---|---|---|---|
| | | | mAP | AP$_{0.50}$ | mAP | AP$_{0.50}$ | mAP | AP$_{0.50}$ |
| ✓ | | | 54.0 | 83.0 | 55.7 | 83.8 | 58.0 | 85.0 |
| | ✓ | | 53.2 | 82.5 | 54.3 | 83.0 | 57.2 | 84.5 |
| | | ✓ | 53.8 | 82.9 | 55.2 | 83.3 | 57.8 | 84.8 |
| ✓ | ✓ | | 54.4 | 83.2 | 56.0 | 83.9 | 58.4 | 85.2 |
| ✓ | | ✓ | 55.7 | 83.5 | 56.7 | 84.2 | 58.7 | 85.5 |
| | ✓ | ✓ | 54.1 | 83.0 | 55.9 | 83.9 | 58.1 | 85.2 |
| ✓ | ✓ | ✓ | **56.1** | **83.7** | **57.1** | **84.3** | **59.0** | **85.5** |

bility. However, the performance gap remains relatively small, particularly for larger model variants. FSNN also achieves higher computational efficient due to its use of ternary convolutions, whereas FANN relies on more computation-intensive standard convolutions.

### 4.4.2 COMPARISON BETWEEN CONVENTIONAL MOE AND THE PROPOSED HE-MOE

Table 4 compares the conventional MoE module (Fig. 1(a)) with the proposed HE-MoE module (Fig. 1(b)). The results clearly demonstrate that HE-MoE consistently outperforms the conventional MoE, and this improvement can be attributed to two key factors. First, HE-MoE introduces multiple enhancement priors, including dehazing, color correction, and contrast enhancement, whereas the conventional MoE operates directly on raw RGB images without any enhancement priors. By leveraging complementary enhancement cues, HE-MoE provides cleaner and more informative feature representations, which significantly boost detection accuracy. Second, HE-MoE adopts a modality-specific expert design, where each expert processes a distinct enhancement modality. This encourages experts to specialize in different enhancement domains, thereby maximizing diversity and enabling the model to exploit complementary information more effectively. In contrast, the conventional MoE feeds the same input representation to all experts, which often leads to homogeneous experts that learn overlapping or redundant features. Without a mechanism to encourage diversity, conventional MoE fails to fully utilize the potential of multiple experts.

### 4.4.3 COMPARISON BETWEEN INDIVIDUAL ENHANCEMENT EXPERTS

To systematically assess the contribution of the three enhancement strategies in the HE-MoESNN framework (color correction, dehazing, and contrast enhancement), we conduct three sets of experiments: (i) applying the framework with a single enhancement prior, (ii) combining any two enhancement priors, and (iii) using the full framework with all three priors.

The results in Table 5 lead to three main observations. First, the enhancement experts are complementary. The full model with all three priors achieves the highest detection performance, highlighting the advantage of jointly addressing multiple types of underwater image degradation. This combination produces richer feature representations and leads to more accurate object localization. Second, dehazing and contrast enhancement contribute more strongly to performance improvements than color correction. This is because underwater images often suffer from severe turbidity and low contrast, which directly degrade visibility and hinder reliable feature extraction. Correcting these degradations provides a more substantial boost to detection robustness. Finally, dehazing is the most critical factor for underwater object detection. By removing haze and restoring scene clar-

ity, it significantly enhances object visibility and preserves fine structural details, forming a strong foundation for downstream detection tasks. Overall, these findings underscore the importance of combining complementary enhancement strategies, particularly dehazing, to achieve robust, generalizable underwater object detection across diverse and challenging environments.

## 5  CONCLUSION

This work introduced HE-MoESNN, a spiking neural network framework that integrates a Mixture of Heterogeneous Enhancement Experts with a lightweight FSNN backbone to address the unique challenges of underwater object detection. By explicitly guiding experts to specialize in dehazing, color correction, and contrast enhancement, HE-MoE generates complementary, domain-aware feature representations that significantly improve detection robustness under severe underwater degradation. Our results on RUOD and DUO show that HE-MoESNN outperforms state-of-the-art detectors in accuracy while maintaining competitive efficiency, striking an excellent balance between precision and computational cost.

In future work, we plan to explore energy-efficient neuromorphic deployment, temporal consistency modeling with event-based data, and adaptive expert selection to further improve robustness and reduce computational cost across diverse underwater scenarios.

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

## A APPENDIX

### A.1 USE OF LLMS

Large Language Models (LLMs) were used solely to assist with writing and polishing the text.

### A.2 CODE OF ETHICS AND ETHICS STATEMENT

The research conducted in the paper conform, in every respect, with the ICLR Code of Ethics https://iclr.cc/public/CodeOfEthics.

### A.3 REPRODUCIBILITY

This paper provides all necessary details to enable reproduction of the main experimental results, including dataset descriptions, training procedures, training parameters, and evaluation protocols.

