# OpenReview forum: "Spiking Neural Network with Mixture of Heterogeneous Enhancement Experts for Robust Underwater Object Detection"
_ICLR.cc/2026/Conference — Submitted to ICLR 2026_

### Official Review · Reviewer_Hp94 · 2025-10-28

**Soundness:** 3
**Presentation:** 3
**Contribution:** 2
**Rating:** 6
**Confidence:** 2

**Summary:**

This paper proposes a spiking neural network framework that combines heterogeneous enhancement experts with a forward spiking neural network backbone for underwater object detection. The HE-MoE module assigns different enhancement modalities to three specialized experts. The FSNN backbone replaces standard convolutions with ternary operations using signed spiking neurons to reduce computational cost. Experiments on RUOD and DUO datasets demonstrate competitive performance with improved efficiency compared to existing methods.

**Strengths:**

1. The heterogeneous expert design that assigns modality-specific inputs to different experts is a meaningful departure from conventional MoE approaches.

2. The paper provides thorough ablation studies examining individual enhancement components, backbone comparisons, and systematic evaluation across multiple model scales.

3. The focus on computational efficiency through spiking neural networks addresses real deployment constraints for underwater detection systems.

**Weaknesses:**

1. The core components (spiking neural networks, MoE architectures, underwater enhancement techniques) are well-established, the main contribution is their combination rather than fundamental algorithmic innovation.

2. The computational cost analysis only considers arithmetic operations while ignoring memory overhead, routing costs, and the fact that all three experts must be loaded simultaneously, potentially making the efficient SNN approach less efficient than claimed in real deployment scenarios.

3. The paper claims computational efficiency as a key advantage but their largest model (HE-MoESNN-L) actually uses more parameters and FLOPs than many baselines.

**Questions:**

1. Why exactly three experts, and how was this number determined? The paper doesn't explain if 2 or 4 experts might work better or provide guidelines for choosing enhancement types.

2. Does the method work beyond these two specific datasets? The authors only test on RUOD and DUO but underwater conditions vary dramatically. What about different water clarity, depths, or lighting?

---

> ### Author Response · Authors · 2025-11-20
> **Thank you to Reviewer Hp94 for the valuable suggestion**
>
> ***Regarding Weakness 1 the main contribution is their combination rather than fundamental algorithmic innovation.***
>
> Thank you for raising this point. We would like to clarify where the novelty of our work lies:
>
> **(1) Novelty of the heterogeneous enhancement-driven MoE (HE-MoE) formulation.**
>
> Prior MoE-based vision models typically employ homogeneous experts—that is, experts with the same architecture, operating on the same modality, and learning similar feature spaces. In contrast, our HE-MoE introduces structurally identical but functionally heterogeneous experts, each designed around a distinct underwater degradation prior: dehazing, color correction, and contrast enhancement.
>
> This heterogeneous design explicitly aligns expert specialization with known physical degradation mechanisms in underwater imaging, which has not been explored in previous MoE architectures.
>
> **(2) MoE integration within an FSNN pipeline—an under-explored direction**
>
> Integrating a mixture-of-experts mechanism into a spiking neural network presents unique challenges not seen in ANN-based MoEs. SNNs rely on: ternary spike activations, sparse firing dynamics, and addition/subtraction-only convolutions. As shown in Table 3 and our expanded energy analysis, the proposed HE-MoE + FSNN combination preserves high detection accuracy while dramatically reducing expensive ANN-style multiplications. This demonstrates that our formulation is both effective and energy-efficient—offering a level of functionality not present in existing underwater detection frameworks.
>
> We hope this clarification helps highlight the conceptual and technical novelty of our approach.
>
> ***Regarding Weaknesses 2 and 3 computational cost analysis***
>
> **Weakness 2 FPS and estimated energy consumption**
>
> Thanks for your valuable suggestion, we have now added FPS and estimated energy consumption results for HE-MoESNN; due to space limitations, these details are provided in our response to Reviewer 5NuK.
>
> **Weakness 3**
>
> We agree that the largest model (HE-MoESNN-L) achieves the best performance but uses more parameters and FLOPs than some baselines. However, HE-MoESNN-S offers a strong balance between accuracy and efficiency. On the DUO dataset, it achieves the second-best accuracy while using far fewer parameters and FLOPs than other models. On the RUOD dataset, HE-MoESNN-S reaches 57.1 mAP, only slightly lower than DJNet (57.5 mAP), yet is much more efficient (8.56M vs. 58.48M parameters and 35.86G vs. 69.51G FLOPs). This demonstrates that HE-MoESNN-S is both accurate and lightweight.
>
> ***Regarding Question 1 Why exactly three experts***
>
> We selected these three enhancement methods because each targets a different type of underwater degradation. Dehazing (DH) reduces backscatter and turbidity, color correction (CC) handles wavelength-dependent color shifts, and contrast enhancement (CE) improves visibility in low-illumination or low-contrast scenes.
>
> These degradations match the main physical factors described in classical underwater imaging theory (e.g., the Jaffe–McGlamery model), so they reflect well-established domain knowledge. Although many deep learning–based enhancement methods (e.g., UIE-Net, Water-Net) exist, they combine multiple enhancement effects, making it difficult to analyze the contribution of each type. Our goal is to evaluate each enhancement prior individually and study how they interact within the HE-MoE framework.
>
> As shown in Section 4.4.3, the three enhancement experts are complementary, and dehazing proves to be the most important component for achieving robust and generalizable underwater detection across diverse environments.
>
>
> ***Regarding Question 2 Generalization beyond RUOD and DUO***
>
> We agree that evaluating generalization across diverse environments is an important concern. This is precisely why we selected RUOD as one of our primary benchmarks. RUOD aggregates images from multiple online sources and datasets (including URPC2020), making it one of the most diverse and comprehensive underwater detection datasets available. It encompasses a wide range of heterogeneous degradations, such as severe color distortion, varying haze levels, and low-contrast scenes, which directly correspond to the types of degradation that HE-MoE is designed to handle.
>
> Given this diversity, RUOD serves as an effective testbed for assessing the generalization capability of underwater detection models.

---

### Official Review · Reviewer_dBZZ · 2025-10-29

**Soundness:** 2
**Presentation:** 3
**Contribution:** 2
**Rating:** 2
**Confidence:** 5

**Summary:**

The authors propose HE-MoESNN, a spiking neural network that integrates a Mixture of Heterogeneous Enhancement Experts (HE-MoE) with a lightweight Forward Spiking Neural Network (FSNN) backbone. Unlike conventional MoE frameworks that feed identical inputs to all experts, HE-MoE assigns modality-specific inputs—including dehazing, color correction, and contrast enhancement—to three parallel experts and fuses their outputs through a shared gating router.

**Strengths:**

1. The manuscript presents a clear and well-structured method.
2. Good performance is demonstrated according to the comparisons reported in the paper.

**Weaknesses:**

1. While the proposed framework is technically well-described, the network design is incremental and lacks theoretical depth. Although the paper combines image enhancement techniques with a mixture-of-experts architecture to design a lightweight network, it lacks novel insight.
2. It is unclear why these three enhancement methods were selected. How do other enhancement methods perform in comparison? Moreover, the paper lacks a thorough analysis of the computational efficiency and enhancement quality of the chosen methods, as well as their impact on the subsequent model. Overall, the work lacks comprehensive justification and analysis.
3. The rationale for selecting YOLOX as the base model is not clearly justified. It would be valuable to analyze how the proposed HE-MoE performs when combined with other representative object detection architectures.

**Questions:**

Please see the Weaknesses.

---

> ### Author Response · Authors · 2025-11-20
> **Thank you to Reviewer dBZZ for the valuable suggestion**
>
> ***Regarding Weakness 1: Lack of novel insight***
>
> Thank you for raising this point. We would like to clarify where the novelty of our work lies:
>
> **(1) Novelty of the heterogeneous enhancement-driven MoE (HE-MoE) formulation.**
>
> Prior MoE-based vision models typically employ homogeneous experts—that is, experts with the same architecture, operating on the same modality, and learning similar feature spaces. In contrast, our HE-MoE introduces structurally identical but functionally heterogeneous experts, each designed around a distinct underwater degradation prior: dehazing, color correction, and contrast enhancement.
>
> This heterogeneous design explicitly aligns expert specialization with known physical degradation mechanisms in underwater imaging, which has not been explored in previous MoE architectures.
>
> **(2) MoE integration within an FSNN pipeline—an under-explored direction**
>
> Integrating a mixture-of-experts mechanism into a spiking neural network presents unique challenges not seen in ANN-based MoEs. SNNs rely on: ternary spike activations, sparse firing dynamics, and addition/subtraction-only convolutions. As shown in Table 3 and our expanded energy analysis, the proposed HE-MoE + FSNN combination preserves high detection accuracy while dramatically reducing expensive ANN-style multiplications. This demonstrates that our formulation is both effective and energy-efficient—offering a level of functionality not present in existing underwater detection frameworks.
>
> We hope this clarification helps highlight the conceptual and technical novelty of our approach.
>
> ***Regarding Weakness 2***
>
> **Unclear why these three enhancement methods were chosen**
>
> We selected these three enhancement methods because each targets a distinct and non-overlapping type of underwater degradation. Specifically: Dehazing (DH) reduces backscatter and turbidity effects; Color correction (CC) addresses wavelength-dependent attenuation, which causes the characteristic blue/green color shift; Contrast enhancement (CE) improves visibility and restores edge information in low-illumination or low-contrast regions.
>
> These three degradations correspond directly to the dominant physical factors described in classical underwater imaging theory (e.g., the Jaffe–McGlamery model), and thus reflect well-established domain knowledge. While many deep learning-based enhancement methods (e.g., UIE-Net, Water-Net) exist, they typically fuse multiple enhancement effects into a single operation. Such methods do not allow us to isolate and analyze the contribution of each degradation type separately. In contrast, our goal in this paper is to evaluate each enhancement prior individually (see Section 4.4.3) and examine how they interact within the HE-MoE framework.
>
> **Lacks a thorough analysis of the computational efficiency and enhancement quality**
>
> *(1) Computational efficiency.*
>
> The three selected enhancement methods are lightweight classical (non–deep learning) algorithms with negligible computational overhead. In the original manuscript, efficiency was evaluated using standard proxies (FLOPs and parameter count). We have now added FPS and estimated energy consumption results for HE-MoESNN; due to space limitations, these details are provided in our response to Reviewer 5NuK.
>
> *(2) Enhancement quality on downsteam detection model.*
>
> Figure 3 currently compares detection results with and without the fusion of all three enhancement strategies, with detections shown only on the original degraded images. In the supplementary material, we will include the corresponding detection results for each individual enhanced modality to more clearly illustrate how each enhancement method contributes to the final detection performance.
>
>
> ***Regarding Weakness 3: Why YOLOX?***
>
> Thank you for raising this point.
>
> **(1) YOLOX is selected for its strong compatibility with SNN/FSNN architectures.**
>
> YOLOX’s decoupled head, anchor-free design, and simpler multi-scale topology make it significantly more stable during ANN→SNN conversion and during FSNN training. In contrast, detectors such as YOLOv5 or CenterNet tend to produce dense activations or heatmap-style outputs that lead to unstable spike distributions and degraded SNN performance. Thus, YOLOX provides a balanced trade-off between model stability, efficiency, and accuracy in the spiking domain.
>
> **(2) Our contribution targets the enhancement–fusion mechanism, not the base detector.**
>
> Our goal is not to propose a new detection architecture, but rather to introduce a modular enhancement-driven, degradation-aware SNN fusion mechanism (HE-MoE). YOLOX serves as a strong, lightweight, and representative testbed, but the proposed HE-MoE is architecture-agnostic and can be integrated into other detectors as well.

---

### Official Review · Reviewer_iD4T · 2025-10-31

**Soundness:** 3
**Presentation:** 2
**Contribution:** 2
**Rating:** 4
**Confidence:** 5

**Summary:**

This paper introduced the HE-MoESNN framework for underwater object detection, targeting the challenges of image degradation caused by light absorption and scattering. The proposed HE-MoE module integrates heterogeneous experts designed for defogging, color correction, and contrast enhancement, and employs a shared gating router to adaptively fuse their outputs. Furthermore, the paper employed a lightweight FSNN backbone to replace traditional convolutional structures, which significantly reduced computational cost while maintaining high accuracy on the RUOD and DUO data sets.

**Strengths:**

1. The HE-MoE module breaks the limitation in traditional MoE where all experts share the same input, enabling experts to specialize in different enhancement domains. The application of FSNN in underwater object detection also exhibits certain innovativeness.
2. The HE-MoE module possesses energy efficiency advantages on embedded or edge underwater platforms
3. The paper included detailed component-wise analyses that verified the contribution of each module (HE-MoE, FSNN,different enhancement modalities, etc.).

**Weaknesses:**

1. The MoE baseline uses a single RGB input, while the proposed HE-MoE receives three enhanced images (dehazed, color-corrected, and contrast-enhanced). Therefore, it remains unclear whether the observed improvement stems from the multi-expert architecture itself or merely from the enhanced image inputs.
2. The paper frequently emphasized the complementarity among the three enhancement experts, but no experimental analysis (e.g., feature similarity or mutual information) was provided to verify whether these experts indeed learn complementary representations.
3. HE-MoESNN-L’s FSNN-FLOPs only include low-energy addition/subtraction, while DJLNet’s ANN-FLOPs involve high-energy multiplication-addition. HE-MoESNN-L has the multi-modal HE-MoE module , and DJLNet only a single-modal decolorization module. It’s unclear if their performance gap (mAP 59.0 vs 57.5) and efficiency difference came from "HE-MoE + FSNN" advantages or inconsistent computational/functional configurations.

**Questions:**

1. The motivation focused on mitigating underwater degradation, yet experiments only showed detection accuracy. Have the authors evaluated whether the model actually improves visual quality or robustness under different degradation levels to substantiate the claimed motivation?
2. Beyond theoretical FLOPs and number of parameters, did you measure real-world inference speed or energy consumption (e.g., FPS, latency, or power usage) to assess deployment feasibility on underwater devices?

---

> ### Author Response · Authors · 2025-11-20
> **Thank you to Reviewer iD4T for the valuable suggestion**
>
> ***1. “MoE and other ANN-based baselines (DJLNet) use a single RGB input, but HE-MoE uses three enhanced images.”***
>
> We appreciate the reviewer’s concern regarding how to disentangle the contribution of the MoE architecture from the contribution of the enhanced inputs.
>
> In section 4.4.1, we conduct the ablation study on the HE-MoE architecture. Specially, we conduct two sets of experiments to evaluate the contribution of the proposed HE-MoE architecture. First, we compare models trained with and without the HE-MoE module to quantify its direct impact on detection accuracy. Second, we examine the HE-MoE architecture using two different backbones—FANN and FSNN—to analyze how the choice of backbone influences the overall performance.
>
> **(1) With the FANN versus FSNN backbones**
>
> Table 3 shows that the HE-MoE architecture paired with the FANN backbone outperforms its counterpart using the FSNN backbone, indicating that FANN provides stronger feature extraction capability. In contrast, FSNN achieves higher computational efficiency due to its use of ternary convolutions, whereas FANN depends on more computation-intensive standard convolutions.
>
> This result supports our claim that the performance and efficiency differences observed between HE-MoE and DJINet (mAP 59.0 vs. 57.5) arise from the combined advantages of HE-MoE + FSNN. Notably, the HE-MoE + FANN configuration performs better than both HE-MoE + FSNN and DJINet, demonstrating that the performance gain is primarily due to the HE-MoE architecture rather than an artifact of the backbone’s computational design.
>
> **(2) Standard MoE vs. HE-MoE under comparable backbones**
>
> In Table 4, we provide a direct comparison between a conventional MoE and our proposed HE-MoE, both receiving the same three enhanced images as inputs. Across all three model scales (Tiny/Small/Large), replacing the standard MoE with HE-MoE consistently yields higher mAP. This demonstrates that the improvements come from the architectural design of HE-MoE rather than merely from supplying enhanced images.
>
> **(2) Contribution of enhancement modalities and evidence of complementarity**
>
> Table 5 further analyzes the enhancement modalities themselves by enabling different subsets of the three experts (DH/CC/CE) while keeping the HE-MoE architecture fixed. The results clearly show that each enhancement prior contributes useful information and that combining them leads to progressively better performance, indicating strong complementarity among the experts.
>
> We thank the reviewer for suggesting more in-depth analyses (e.g., feature similarity or mutual information) to more explicitly verify complementarity. At this stage, we are unable to implement such analyses due to limited resources and time, but we acknowledge this as valuable future work and will mention it in the revised manuscript.
>
>
> ***2. “Motivation focuses on mitigating underwater degradation, yet experiments only show detection accuracy.”***
>
> We agree that explicitly linking this to visual quality and degradation severity is important. Qualitative evidence already included. Figure 3 compares detection results with and without the HE-MoE module under severe color distortion, heavy haze, and low-contrast conditions. As shown in the figure, in heavily degraded scenes, the model without HE-MoE often misses objects or produces incorrect localizations. In contrast, the model with HE-MoE recovers clearer object boundaries and yields correct detections, demonstrating improved robustness to degradations.
>
> For space considerations, Figure 3 currently displays only the original degraded images. If required, we will include the three corresponding enhanced images (dehazed, color-corrected, and contrast-enhanced) in the supplementary material to further illustrate how the enhancement modalities contribute to improved detection.
>
> ***3. FPS and energy consumption.***
>
> We fully agree that FPS, latency, and energy consumption warrant clearer discussion. We have provided the FPS and estimated energy consumption results for HE-MoESNN. Due to space limitations here, we kindly refer the reviewer to our detailed response to Reviewer 5NuK, where we report and discuss FPS and energy consumption comprehensively.
>
> | Model        | FPS   | Time Step ($T$) | Firing Rate ($fr$) | FLOPs (G) | Energy ($E$)             |
> |--------------|-------|------------------|----------------------|-----------|---------------------------|
> | ERLNet (ANN) | –     | –                | –                    | 54.71     | $684\ \text{mJ}$         |
> | HE-MoESNN-T  | 93.8  | 1                | 0.2388               | 22.28     | $0.410\ \text{mJ}$        |
> | HE-MoESNN-S  | 85.6  | 1                | 0.2190               | 35.86     | $0.605\ \text{mJ}$        |
> | HE-MoESNN-L  | 40.3  | 1                | 0.1806               | 170.03    | $2.364\ \text{mJ}$        |

---

### Official Review · Reviewer_5NuK · 2025-11-02

**Soundness:** 2
**Presentation:** 2
**Contribution:** 2
**Rating:** 6
**Confidence:** 4

**Summary:**

This paper introduces HE-MoESNN, a spiking neural network model for underwater object detection. The key
idea is to use a mixture of enhancement experts, where each expert focuses on a different type of underwater
image improvement (one for dehazing, one for color correction and one for contrast enhancement). A routing
module combines the outputs from these experts. The detection network is built as a Forward Spiking Neural
Network with signed spiking neurons and ternary convolutions to reduce computation. The model is evaluated
on the RUOD and DUO datasets and shows higher detection accuracy and better efficiency compared to both
standard deep neural networks and previous SNN-based methods.

**Strengths:**

The motivation of the paper is clear and makes sense for the underwater setting. The authors explain that underwater images often suffer from haze, color distortion and low contrast and the enhancement modules they choose directly target these issues.

The idea of using a heterogeneous mixture-of-experts is meaningful. Since each expert receives a differently enhanced version of the image, this encourages each branch to learn different features instead of all doing the same thing.

The use of a Forward Spiking Neural Network with signed spiking neurons and ternary convolutions helps reduce computation. This is useful for situations where power or hardware resources are limited, such as underwater robots or remote-operated devices

The experiments are broad and include comparisons with general object detectors, underwater-focused models and also previous spiking neural network detectors. The proposed model shows improvements in both detection accuracy and computational efficiency.

The ablation studies are helpful because they show the effect of each key component separately, including the heterogeneous experts module, the MoE vs standard fusion and the difference between the spiking and non-spiking backbone.

**Weaknesses:**

The paper mentions that the model is efficient due to the spiking design and ternary convolutions, but there are no actual hardware tests. There are no latency, FPS or energy measurements, so it is hard to know how the model performs in real deployment, especially
since SNN efficiency can depend on the device used.

The results seem to come from single runs and there are no standard deviation or confidence intervals reported. Because of this, it is difficult to know how stable the improvements are or whether they might vary with different random seeds.

The experiments are only done on two underwater datasets (RUOD and DUO). Both datasets are from similar underwater environments. So, it is unclear how well the model would perform in other settings, such as different water types, lighting, turbidity or general degraded images.

Some implementation details are not described fully, such as how the enhancement modules are parameterized or how the routing and spiking thresholds are tuned. This may make it harder for others to reproduce the exact performance.

**Questions:**

The paper mentions that the model is efficient due to the spiking design and ternary convolutions, but there are no actual hardware tests. There are no latency, FPS or energy measurements, so it is hard to know how the model performs in real deployment, especially
since SNN efficiency can depend on the device used.

The results seem to come from single runs and there are no standard deviation or confidence intervals reported. Because of this, it is difficult to know how stable the improvements are or whether they might vary with different random seeds.

The experiments are only done on two underwater datasets (RUOD and DUO). Both datasets are from similar underwater environments. So, it is unclear how well the model would perform in other settings, such as different water types, lighting, turbidity or general degraded images.

Some implementation details are not described fully, such as how the enhancement modules are parameterized or how the routing and spiking thresholds are tuned. This may make it harder for others to reproduce the exact performance.

---

> ### Author Response · Authors · 2025-11-20
> **Thank you to Reviewer 5NuK for the valuable suggestion**
>
> ***1. FPS and energy consumption.***
>
> Good suggestions, we have added FPS and energy consumption of our HE-MoESNN models.
>
> **FPS measurement:**
> All FPS values were obtained on a workstation equipped with an Intel(R) Xeon(R) Silver 4114 CPU (2.20 GHz) and a Tesla V100 GPU (32 GB).
>
> We report FPS only for our HE-MoESNN models and do not provide FPS for the other comparison methods because the reported results on RUOD (Table 1) and DUO (Table 2) come directly from the official or previous papers. Most of these previous works did not report FPS, so we reported their results as originally presented.
>
> | Model        | FPS   | Time Step ($T$) | Firing Rate ($fr$) | FLOPs (G) | Energy ($E$)             |
> |--------------|-------|------------------|----------------------|-----------|---------------------------|
> | ERLNet (ANN) | –     | –                | –                    | 54.71     | $684\ \text{mJ}$         |
> | HE-MoESNN-T  | 93.8  | 1                | 0.2388               | 22.28     | $0.410\ \text{mJ}$        |
> | HE-MoESNN-S  | 85.6  | 1                | 0.2190               | 35.86     | $0.605\ \text{mJ}$        |
> | HE-MoESNN-L  | 40.3  | 1                | 0.1806               | 170.03    | $2.364\ \text{mJ}$        |
>
> Moreover, in our original manuscript, our efficiency claims are based on standard computational proxies (FLOPs and parameters). Tables 1–2 and Fig. 2 report FLOPs and parameter counts for all baselines. For example, on DUO, HE-MoESNN-S achieves higher mAP than SpikingYOLOX while using far fewer FLOPs (35.86G vs. 151.69G) and parameters (8.56M vs. 49.53M). These metrics are widely used as hardware-independent proxies for runtime and computational cost, especially when results cannot be tested across multiple neuromorphic platforms.
>
> **Energy estimation:**
> Precise measurement of energy consumption on real neuromorphic or embedded hardware is difficult due to platform-dependent variations. Following the standard practice in previous studies [Ref1, Ref2], we estimate energy usage by assuming $12.5\ \mathrm{pJ}$ per FLOP and $77\ \mathrm{fJ}$ per synaptic operation (SOP).
>
> $E_{\text{SNN}} = 77\text{fJ} \times  fr \times T \times \text{FLOPs}$;
>
> $E_{\text{ANN}} = 12.5\text{pJ} \times \text{FLOPs}$.
>
> As shown in the table above, ERLNet (54.71G FLOPs) is the most energy-efficient ANN baseline, requiring $684\ \mathrm{mJ}$ per inference. In contrast, our HE-MoESNN models achieve substantially lower estimated energy consumption.
>
> [Ref 1] Yufei Guo et al,. Ternary spike: Learning ternary spikes for spiking neural networks. In Proceedings of the AAAI conference on artificial intelligence, pages 12244–12252, 2024. 5
>
> [Ref 2] Tianyu Song et al,. Learning a spiking neural network for efficient image deraining. 2024. 5
>
> ***2. Single runs and lack of variance/confidence intervals***
>
> We agree that reporting variability is important for evaluating the stability of performance gains. Our main tables (Table 1 and Table 2) report a single run per method primarily to remain consistent with prior underwater detection works. For several comparison baselines, the reported results are taken directly from their official implementations or previously published papers, and we do not retrain those models ourselves.
>
> Furthermore, due to computational constraints, we are currently unable to perform multi-seed experiments. Each model variant requires 300–500 training epochs on RUOD/DUO, making repeated runs prohibitively expensive within our available resources.
>
> ***3. Generalization beyond RUOD and DUO***
>
> We agree that evaluating generalization across diverse environments is an important concern. This is precisely why we selected RUOD as one of our primary benchmarks. RUOD aggregates images from multiple online sources and datasets (including URPC2020), making it one of the most diverse and comprehensive underwater detection datasets available. It encompasses a wide range of heterogeneous degradations, such as severe color distortion, varying haze levels, and low-contrast scenes, which directly correspond to the types of degradation that HE-MoE is designed to handle.
>
> Given this diversity, RUOD serves as an effective testbed for assessing the generalization capability of underwater detection models.
>
> ***4. Implementation details***
>
> Each expert is instantiated by integrating a standard enhancement method from the literature: Meng et al. (2013) for dehazing, Jha & Bhandari (2024) for color correction, and Zhang et al. (2023) for contrast enhancement. In our implementation, we directly follow the parameter settings recommended in the original papers and use the default configurations provided in their official source code.
>
> For the routing/gating mechanism, we adopt softmax-based routing rather than top-k routing because all three enhancement experts are necessary and complementary. Softmax allows the model to incorporate contributions from all three experts, leading to better overall performance.

---

> ### Comment · Reviewer_5NuK · 2025-11-27
> **Response to Author**
>
> While the authors present the mixture of heterogeneous enhancement experts as the core novelty of the work, the experts themselves are simply a collection of existing enhancement algorithms rather than purpose-built methods tailored for underwater imaging. As a result, the proposed MoE module functions primarily as an aggregation layer over pre-existing techniques, and the manuscript does not offer a clear rationale for why these particular experts were chosen. Underwater images suffer from diverse and complex degradations such as haze and backscatter, color shift, contrast loss, illumination attenuation, and turbidity, yet the selection of experts is not justified in relation to these degradation categories.
>
> The literature contains many competing algorithms for each enhancement modality, including multiple dehazing variants, several color correction pipelines, and diverse contrast enhancement techniques. However, the authors do not compare alternative candidate methods within each category, making it unclear whether the selected experts are optimal or simply convenient. Without such an analysis, confidence in the reported MoE performance is weakened.
>
> Since object detection is performed after the enhancement stage, detection accuracy depends strongly on the quality and suitability of the enhancement applied. The manuscript does not investigate how different enhancement strategies influence the downstream detection task, nor does it compare detection results across alternative enhancement baselines. This gap makes it difficult to interpret whether the proposed MoE configuration is truly the most effective or reliable choice for improving detection performance.
>
> In summary, the study does not provide a systematic examination of alternative enhancement strategies and does not convincingly demonstrate that the chosen experts or the resulting MoE design represent the best possible configuration for underwater detection improvement.

---

> > ### Author Response · Authors · 2025-11-28
> > **Thank you very much for your prompt and insightful suggestions. They helped us clarify important points we previously overlooked.**
> >
> > Thank you very much for your prompt and insightful suggestions. Your comments are very helpful in guiding us to clarify several important ideas that we had previously overlooked.
> >
> > ***The main question you raised is why these three enhancement methods were chosen.***
> >
> > We appreciate the reviewer’s concern regarding this question, as it is essential for clarifying the motivation behind our selection of the three enhancement methods, and we acknowledge that we had previously overlooked providing this explanation in sufficient detail. Specifically, we selected them based on two key considerations:  (1) domain knowledge grounded in classical underwater imaging theory, which identifies the primary and distinct forms of underwater image degradation; and  (2) computational efficiency, as the chosen methods are lightweight and introduce minimal overhead compared to deep learning--based alternatives.
> >
> >
> > **(1) Domain knowledge grounded in classical underwater imaging theory**
> >
> > We selected these three specific enhancement methods because each addresses a distinct and non-overlapping type of underwater image degradation. Specifically, Dehazing (DH)} reduces backscatter and turbidity effects, Color Correction (CC)} compensates for wavelength-dependent attenuation, which leads to the characteristic blue/green color shift. Contrast Enhancement (CE) improves visibility and restores edge information in low-illumination or low-contrast regions. These three degradations correspond directly to the key physical factors described in classical underwater imaging theory (e.g., the Jaffe--McGlamery model), and therefore represent well-established domain knowledge.
> >
> > While many deep learning--based enhancement methods (e.g., UIE-Net, Water-Net) exist, they typically integrate multiple enhancement effects into a single unified model. Such approaches do not allow us to isolate and analyze the contribution of each degradation type separately. In contrast, our study aims to evaluate each enhancement prior individually (see Section~4.4.3) and to investigate how they interact within the HE-MoE framework.
> >
> > **(2) Computational efficiency considerations.**
> >
> > We fully agree with the reviewer that numerous alternative algorithms exist for each enhancement modality. However, most advanced methods are deep learning-based and therefore require substantial training and inference time. Although they often combine multiple enhancement effects, they introduce considerable computational overhead.
> >
> > In contrast, the three selected enhancement methods are lightweight classical (non--deep learning) algorithms. Their computational cost is negligible and they can be seamlessly integrated into our pipeline with minimal overhead, making them more suitable for an analysis-focused study.
> >
> > **Finally**, we sincerely thank you once again for your thoughtful and constructive feedback, which will help us further improve the quality of this paper. We also truly appreciate your time and effort in reviewing our work.

---

### Meta-Review · Area_Chair_QuwV · 2026-01-07

**Summary:**

This paper proposes HE-MoESNN, combining heterogeneous enhancement-driven MoE with a forward spiking neural network for underwater object detection. Reviewers generally agree that the motivation is reasonable and the system is carefully engineered, with solid empirical results and efficiency-oriented design. However, concerns remain regarding the depth of novelty, the necessity/optimality of expert choices, and the extent to which gains stem from architectural insight versus well-integrated components. This is a competent and well-executed submission, but some reviewers felt it falls short of the current acceptance standard. Thus, this paper is rejected.

**Reviewer Concerns:**

The rebuttal largely addressed Reviewer 5NuK’s concerns on efficiency evaluation, implementation details, and practical feasibility, and partially addressed Reviewer iD4T’s questions through additional ablations clarifying the role of HE-MoE versus enhanced inputs. However, concerns raised by Reviewer dBZZ regarding limited conceptual novelty, and by Reviewers iD4T and Hp94 regarding deeper evidence of expert complementarity, optimal expert selection, and broader generalization, remain only partially resolved.

**Reviewer Scores:**

Reviewer 5NuK would likely keep the score around 6 or slightly lower, as the rebuttal clarifies efficiency-related issues but leaves concerns about expert choice and overall novelty. Reviewer iD4T might reasonably increase the score from 4 to around 5, given the added ablations and the clearer separation between the effects of HE-MoE and enhanced inputs. Reviewer dBZZ is unlikely to change substantially, with at most a shift from 2 to 3, since the core concern about limited novelty remains. Reviewer Hp94 would likely stay near 6, with a possible small positive adjustment after the clarifications.

Overall, the post-rebuttal scores would remain mixed. The rebuttal helped resolve several practical and fairness-related concerns, leading to a likely upward adjustment from some borderline reviewers. At the same time, concerns about conceptual novelty, expert optimality, and deeper evidence of complementarity were only partially addressed.

---

### Decision · Program_Chairs · 2026-01-26

Reject